# Gold-catalyzed four-component multi-functionalization of alkynes

Shangwen Fang[1], Jie Han[1], Chengjian Zhu [1,2], Weipeng Li [1] & Jin Xie [1,3] ✉

The alkyne unit is a versatile building block in organic synthesis and the development of selective multifunctionalization of alkynes is an important object of research in this field. Herein, we report an interesting gold-catalyzed, four-component reaction that achieves the oxo-arylfluorination or oxo-arylalkenylation of internal aromatic or aliphatic alkynes, efficiently breaking a carbon-carbon triple bond and forming four new chemical bonds. The reaction divergence can be controlled by site-directing functional groups in the alkynes; the presence of a phosphonate unit favors the oxo-arylfluorination, while the carboxylate motif benefits oxo-arylalkenylation. This reaction is enabled by an Au(I)/Au(III) redox coupling process using Selectfluor as both an oxidant and a fluorinating reagent. A wide range of structurally diverse α,α-disubstituted ketones, and tri- or tetra-substituted unsaturated ketones have been prepared in synthetically valuable yields and with excellent chemo-, regio- and stereoselectivity. The gram-scale preparation and late-stage application of complex alkynes have further enhanced their synthetic value.

The alkyne unit, an essential skeleton in organic chemistry, is a versatile building block that is widely used in modern organic synthesis[1,2]. There has been increasing focus on the synthesis and transformations of alkynes, such as the Sonogashira reaction and reactions in click chemistry[3–6]. In the past decade, transition metal-catalyzed selective functionalization of alkynes has been shown to provide an efficient platform for diverse syntheses[7–14]. To the best of our knowledge, the difunctionalization of alkynes has progressed considerably while multifunctionalization reactions, forming 4 new bonds remain highly undeveloped (Fig. 1a). In general, the regioselectivity in alkyne functionalization can be well understood in most cases of terminal and conjugated alkynes but it constitutes the most challenging issue for otherwise unbiased internal alkynes[15], in which the directing group has been used increasingly in attempts to improve the regioselectivity of the reaction[16–22].

In recent years, homogeneous gold-catalyzed oxidative functionalization of alkynes, with their excellent π-acidity reactivity[23–52] has been used as a useful strategy for the construction of complex skeletons. However, strategies based on highly regioselective multifunctionalization of internal alkynes are not popular, possibly because of the high redox potential of the Au[I]/Au[III] couple together with the weak interaction between a ligand skeleton and its nucleophilic partners[53]. In addition to the regioselectivity issue, the challenges to gold-catalyzed oxidative multifunctionalization include matching the reaction rates of the oxidation process, nucleophilic addition, and reductive elimination. If any of these three reactions is faster or slower, the regioselective multifunctionalization of internal alkynes[54] will become incapable. Elegant work by Shi et al. in 2021, revealed a three-component di- and trifunctionalization example by using a ketone group as a site-directing group with aromatic diazonium salts (Fig. 1c)[55]. Inspired by this finding, we questioned if we could use phosphonate or carboxylate[49] as a traceless site-directing group that can directly interact with alkynes, allowing for gold-catalyzed regioselective multifunctionalization of internal alkynes.

In this paper, we report our development of a gold-catalyzed, four-component relay multifunctionalization of both aromatic and aliphatic alkynes. This reaction efficiently breaks a carbon-carbon

[1]State Key Laboratory of Coordination Chemistry, Jiangsu Key Laboratory of Advanced Organic Materials, Chemistry and Biomedicine Innovation Center (ChemBIC), School of Chemistry and Chemical Engineering, Nanjing University, 210023 Nanjing, China. [2]Green Catalysis Center, and College of Chemistry, Zhengzhou University, 450001 Zhengzhou, Henan, China. [3]State Key Laboratory of Chemistry and Utilization of Carbon Based Energy Resources, College of Chemistry, Xinjiang University, 830017 Urumqi, China. ✉e-mail: xie@nju.edu.cn

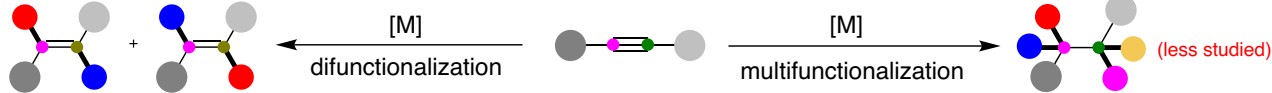

**a** General catalytic difunctionalization and multifunctionalization of alkynes (challenges: diversity and regioselectivity)

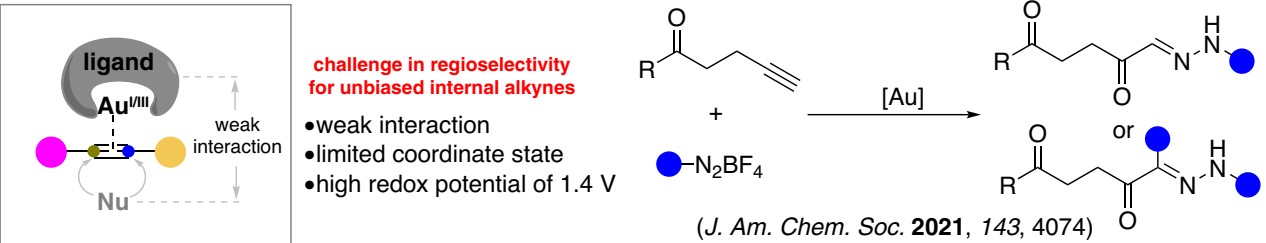

**b** General model for gold-catalyzed alkyne functionalization

**c** Shi's recent work for gold-catalyzed directed strategy

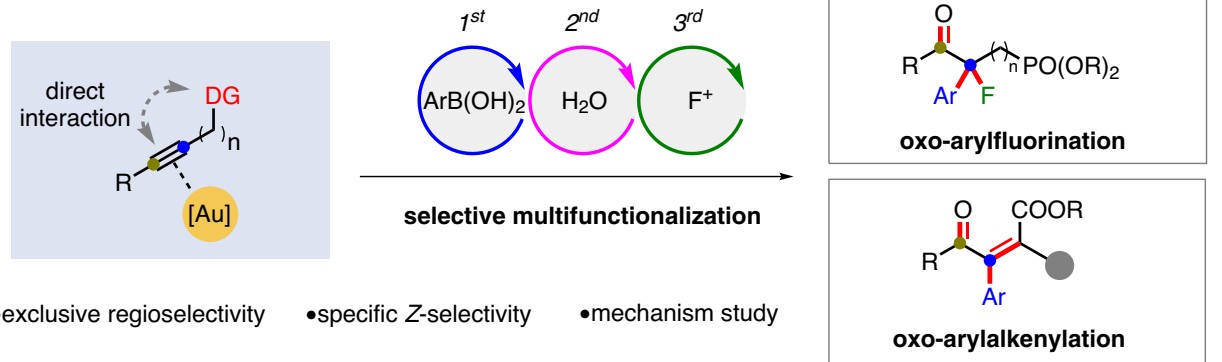

**d This work**: gold-catalyzed multifunctionalization of alkynes tuned by site-directing groups

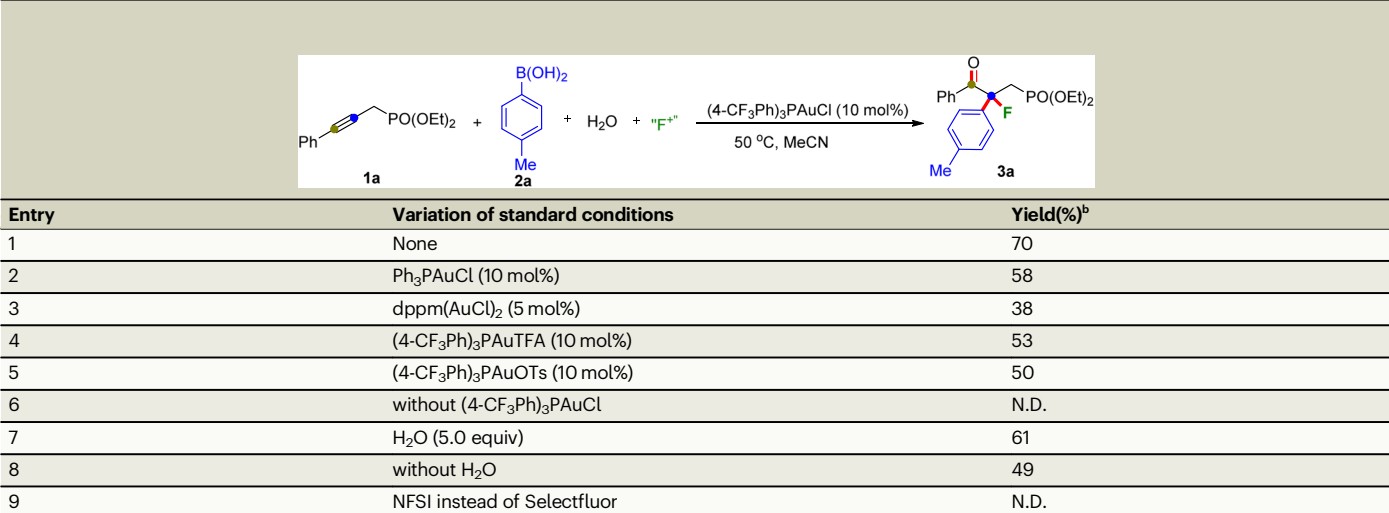

Fig. 1 | **Gold-catalyzed selective multifunctionalization of alkynes. a** General catalytic difunctionalization and multifunctionalization of alkynes. **b** General model for gold-catalyzed alkyne functionalization. **c** Shi's recent work for gold-catalyzed directed strategy. **d** This work: gold-catalyzed multifunctionalization of alkynes tuned by site-directing groups.

## Table 1 | Optimization of gold-catalyzed oxo-arylfluorination[a]

| Entry | Variation of standard conditions | Yield(%)[b] |
|---|---|---|
| 1 | None | 70 |
| 2 | Ph₃PAuCl (10 mol%) | 58 |
| 3 | dppm(AuCl)₂ (5 mol%) | 38 |
| 4 | (4-CF₃Ph)₃PAuTFA (10 mol%) | 53 |
| 5 | (4-CF₃Ph)₃PAuOTs (10 mol%) | 50 |
| 6 | without (4-CF₃Ph)₃PAuCl | N.D. |
| 7 | H₂O (5.0 equiv) | 61 |
| 8 | without H₂O | 49 |
| 9 | NFSI instead of Selectfluor | N.D. |

[a]Standard reaction conditions: **1a** (0.1 mmol), **2a** (0.3 mmol), (4-CF₃Ph)₃PAuCl (10 mol%), Selectflour (4.0 equiv.), H₂O (2.0 equiv.), MeCN (2.0 mL), 50 °C, air, 12 h.
[b]Isolated yields. *dppm* bis(diphenylphosphino)methane, *Selectfluor* 1-chloromethyl-4-fluoro-1,4-diazoniabicyclo[2.2.2]octane bis(tetrafluoroborate), *NFSI* N-fluorobenzenesulfonimide.

triple bond, creating four new chemical bonds and enabling the concise synthesis of highly functionalized ketones (Fig. 1d). The synthetic scope of the reaction can be tuned by the site-directing unit. A phosphonate can benefit the synthesis of α,α-disubstituted ketones by oxo-arylfluorination, while a carboxylate favors the synthesis of tri- and tetra-substituted unsaturated ketones via oxo-arylalkenylation[56]. Detailed studies of the mechanism illustrate the origin of the regioselectivity and can account for the

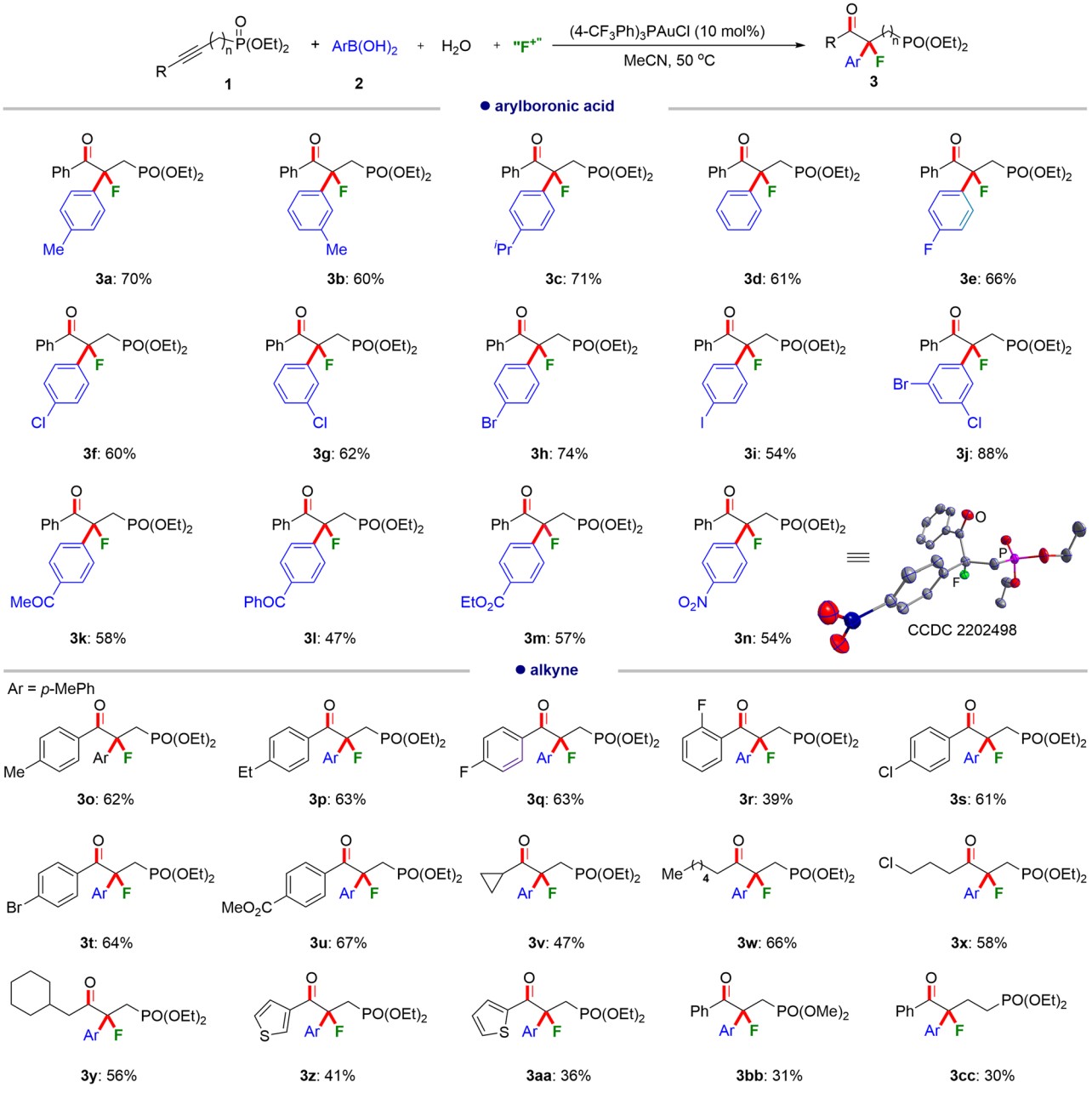

**Fig. 2 | The scope of oxo-arylfluorination of alkynes.** Reaction conditions: (4-CF₃Ph)₃PAuCl (10 mol%), **1** (0.1 mmol, 1.0 equiv.), **2** (0.3 mmol, 3.0 equiv.), Selectfluor (0.4 mmol, 4.0 equiv.), H₂O (0.2 mmol, 2.0 equiv.), MeCN (2.0 mL), air atmosphere, 50 °C, isolated yields are given.

multifunctionalization pathway of alkynes with the different site-directing groups.

## Results

### Reaction optimization

To reveal the direct interaction between gold catalyst and site-directing auxiliary, several kinds of site-directing candidates were investigated (see Supplementary Table 1 for details). The model reaction of an alkyne, diethyl (3-phenylprop-2-yn-1-yl)phosphonate (**1a**), arylboronic acids (**2a**), H₂O and F⁺ reagents were selected to optimize the reaction conditions (Table 1). The standard reaction conditions include using 10 mol% (4-CF₃Ph)₃PAuCl as catalyst, Selectfluor as both an oxidant and a fluorine donor, H₂O as an oxygen-transfer reagent and MeCN as the solvent at 50 °C under an air atmosphere. The reaction with these entities afforded the expected oxo-arylfluorination product, diethyl (2-fluoro-3-oxo-3-phenyl-2-(p-tolyl)propyl)phosphonate (**3a**) in 70% yield (Entry 1). In this reaction, the carbon-carbon triple bond was directly converted into four new chemical bonds, a C=O bond, one C–C bond, and one C-F bond. The screening of other gold catalysts failed to improve further the reaction yields (Entries 2-5). Control experiments showed that this reaction does not occur in the absence of a gold catalyst (Entry 6). The use of 5 equiv. of water resulted in a slightly lower yield of 61% (Entry 7). Interestingly, in the absence of water, the desired product (**3a**) can still be obtained in a yield of 49%, and it was assumed that commercially available aromatic boronic acids contain some water (Entry 8)[57–59]. The Selectfluor was found to be the best choice of oxidant and fluorinating reagent because the use of NFSI in place of Selectfluor resulted in no reaction (Entry 9). We speculated that the Selectfluor could facilitate the rapid change of the oxidation state of the gold species[60,61].

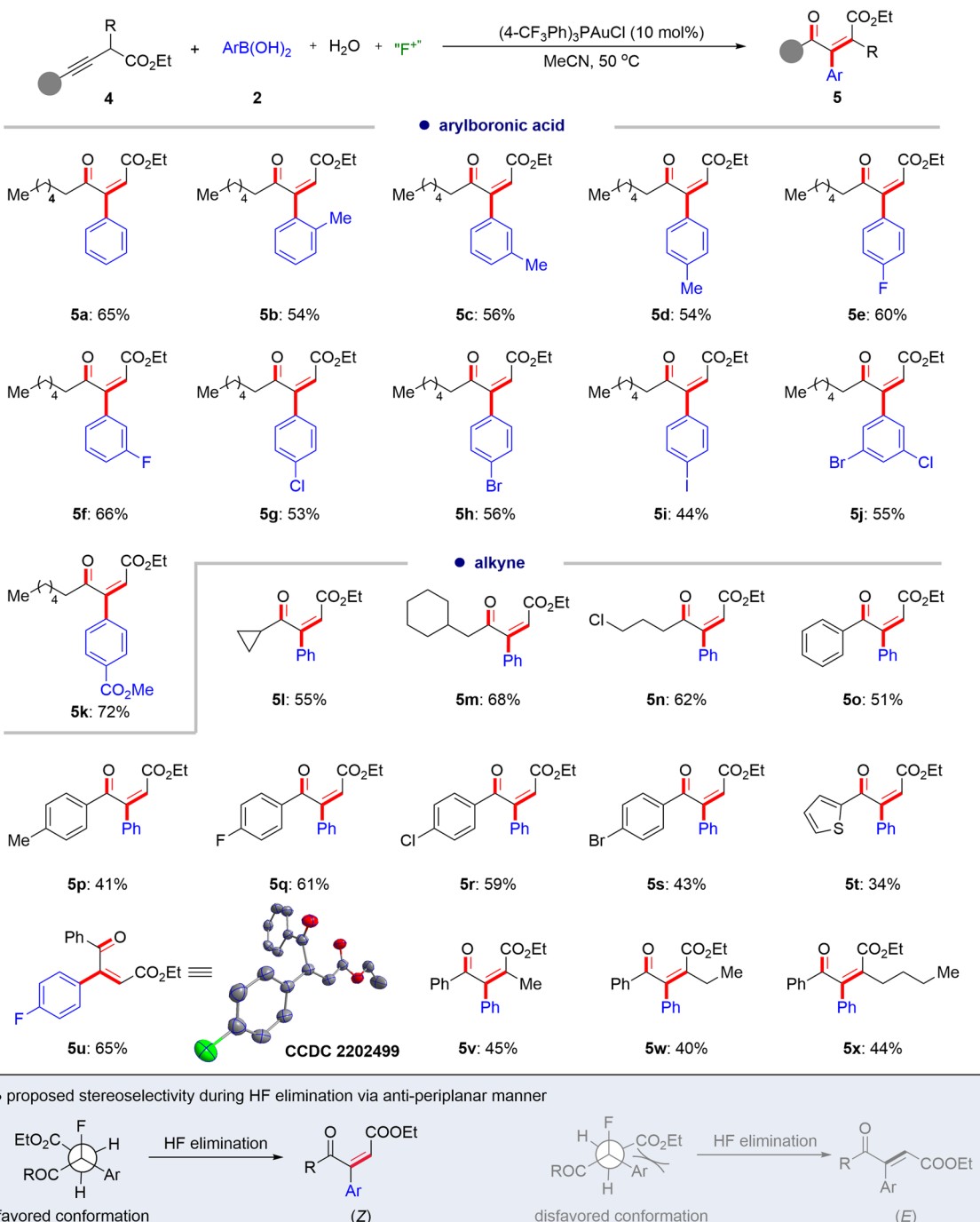

**Fig. 3 | Reaction scope for oxy-arylalkenylation of internal alkynes.** Reaction conditions: **4** (0.1 mmol, 1.0 equiv.), **2** (0.3 mmol, 3.0 equiv.), (4-CF$_3$Ph)$_3$PAuCl (10 mol%), Selectfluor (0.25 mmol, 2.5 equiv.), H$_2$O (0.2 mmol, 2.0 equiv.), MeCN (2.0 mL), air atmosphere, 50 °C, isolated yields are given.

## Substrate scope

With the established reaction conditions in hand, we investigated the scope of the gold-catalyzed oxo-arylfluorination of internal alkynes. As shown in Fig. 2, a wide range of substituted arylboronic acids are competent coupling partners, furnishing the desired products (**3a-3n**) in yields of 47–88%. In general, the alkyl- (**3a-3c**) and halo- (**3e-3j**) substituted arylboronic acids tolerate the reaction conditions well. Notably, the substituents on the *ortho*-position of the phenyl rings such as 2-tolylboronic acid, make the oxo-arylfluorination process reluctant, probably due to the steric influence. Besides, it is found that when the strongly electron-donating 4-methoxy substituted phenylboronic acid was subjected to the standard reaction conditions, no

expected reaction occurred. Furthermore, the versatile functional groups, such as an ester group on the phenyl rings have little influence on the reaction efficiency (**3k-3m**). Importantly, the strongly electron-withdrawing nitro-substituted arylboronic acids are compatible with the reaction, giving rise to the desired products (**3n**) in 54% yield. The structure of **3n** was further confirmed by the X-ray crystallography (CCDC number: 2202498).

Subsequently, we applied this oxo-arylfluorination to a series of different internal alkynes bearing phosphonate as the site-directing group using 4-tolylboronic acid as the coupling substrate. This protocol generally retains the site-exclusive regioselectivity and good functional group compatibility. It was found that both aromatic and

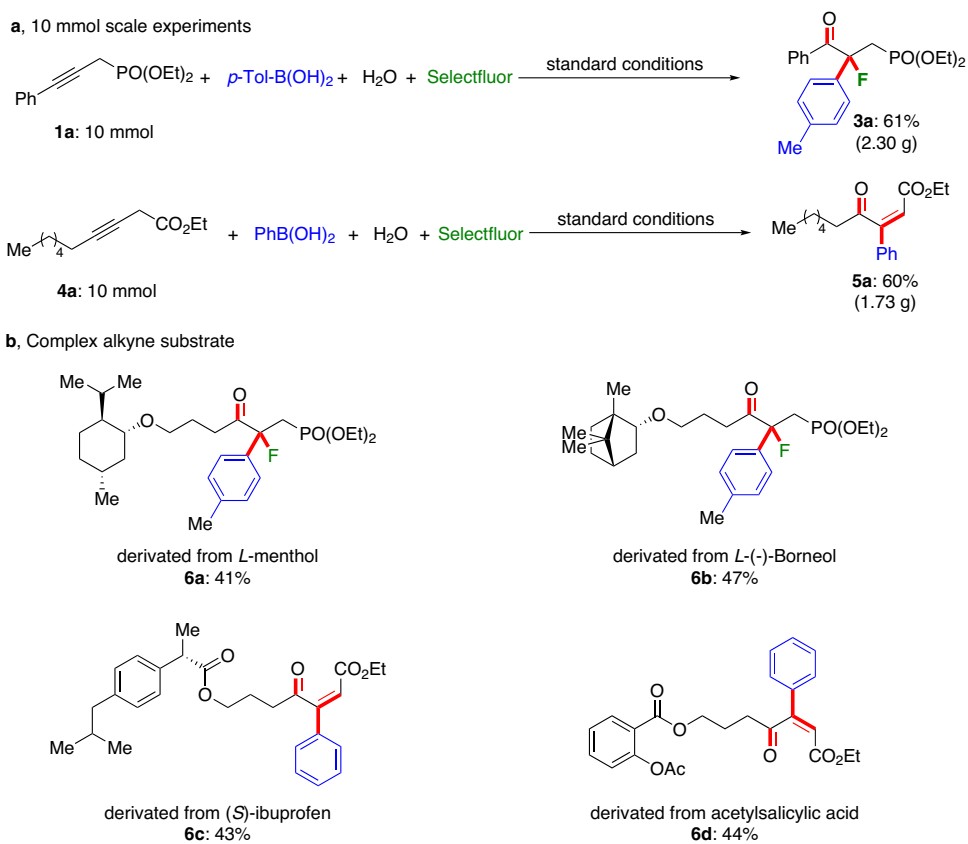

**Fig. 4 | Synthetic application. a** 10 mmol scale-up experiments. **b** Complex alkyne substrate transformations and please see Supplementary Information for reaction conditions.

aliphatic internal alkynes can undergo this regioselective oxo-arylfluorination well, and deliver the final products (**3o-3y**) in synthetically valuable yields. The thiophene-substituted alkynes (**3z** and **3aa**) are also reactive, showing that the reaction is not influenced by the introduction of coordinating sulfur atoms. Furthermore, the broad utility of this method with other phosphonate derivatives was demonstrated. They afforded the desired products (**3bb, 3cc**) in moderate yields.

Interestingly, when the site-directing group was changed from phosphonate to carboxylate, a new oxo-arylalkenylation reaction as opposed to oxo-arylfluorination can occur. This suggests that the reaction divergence can be tuned by the use of different site-directing auxiliaries. The phosphonates favor the oxo-arylfluorination reaction, while the carboxylate can favor the oxo-arylalkenylation reaction. It was found that the optimized reaction conditions for the oxo-arylalkenylation reaction of alkynoate (**4a**) and arylboric acid (**2d**) include the use of 10 mol% (4-CF$_3$Ph)$_3$PAuCl as a catalyst and with a decreased amount of 2.5 equiv. Selectfluor (see Supplementary Information for detailed optimization conditions), which can give the expected oxo-arylalkenylation product (**5a**) in 65% yield.

As illustrated in Fig. 3, a dozen arylboronic acids can proceed well in this oxo-arylalkenylation reaction, furnishing the trisubstituted alkenes (**5a-5k**) in 44–72% yields with only a *Z*-configuration. The substituents on the *ortho-*, *meta-* or *para*-position of the phenyl rings have little influence on the oxo-arylalkenylation process (**5b-5d**). In addition, halogens such as -F, -Cl, -Br, and -I, are tolerated well in the reaction (**5e-5j**), and these functional groups are robust coupling sites in transition metal-catalyzed coupling reactions. Importantly, similar to the oxo-arylfluorination reaction, with the introduction of a site-directing auxiliary, both the aromatic and aliphatic internal alkynes can regioselectively undergo oxo-arylalkenylation reaction smoothly and

in moderate yields (**5l-5u**). Besides the synthesis of trisubstituted alkenes, this protocol is also available for the construction of all-carbon tetra-substituted alkenes (**5v-5x**) in acceptable isolated yields with exclusive stereoselectivity. The molecular structure of oxo-arylalkenylation product (**5u**) was also confirmed by X-ray crystallography (CCDC number: 2202499). We posited that the (*Z*)-configuration of products (**5**) might originate from the H-F elimination with an anti-periplanar staggered conformation, where the favored conformation meets less steric hindrance.

## Synthetic application

To illustrate its synthetic robustness, the 10 mmol scale four-component experiments of both oxo-arylfluorination and oxo-arylalkenylation reactions were performed (Fig. 4a). Under the optimized reaction conditions, a 61% yield of oxo-arylfluorination product (2.30 g) and a 60% yield of the oxo-arylalkenylation product (1.73 g) were obtained. In addition, we examined several complex internal alkynes. We found that they are suitable substrates and can be oxo-arylfluorinated and oxo-arylalkenylated smoothly, affording the desired products (**6a-6d**) in moderate yields (Fig. 4b). This suggests that this site-directing strategy is a practical strategy for gold-catalyzed regioselective multifunctionalization of internal alkynes.

## Mechanism proposal

To gain insight into the mechanism, several control experiments were carried out and are shown in Fig. 5. To determine the source of the oxygen in the newly generated C=O bond derived from the alkyne, systematic labeling experiments were performed for the reaction of an internal alkyne (**1a** or **2a**), tri(4-tolyl)boroxine, H$_2^{18}$O and Selectfluor (Fig. 5a). Notably, along with the increase of the loading of H$_2^{18}$O from 2 equiv. to 10 equiv., the yield of $^{18}$O labeled products (**3a-$^{18}$O** and **5a-$^{18}$O**)

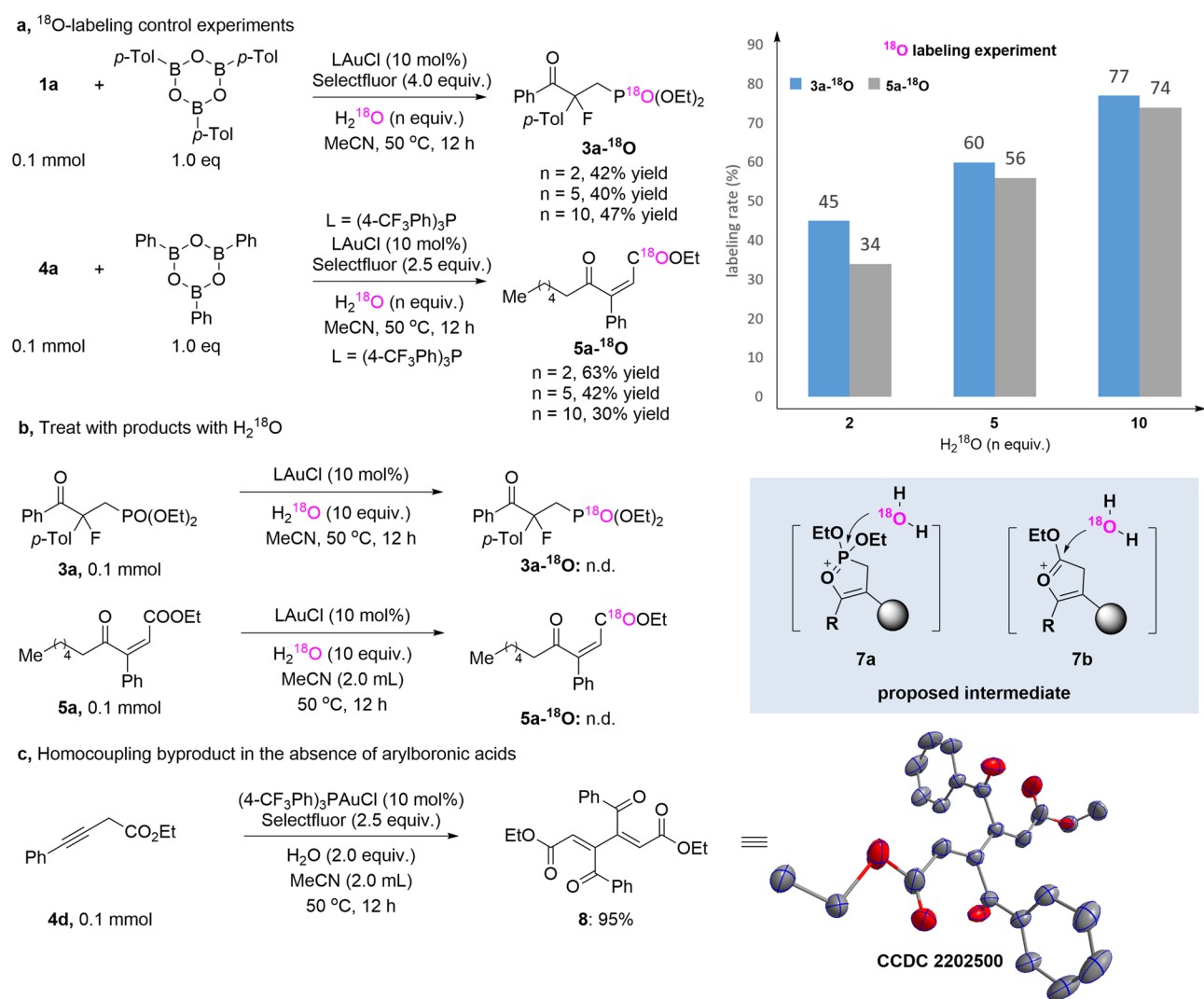

**Fig. 5 | Studies of the mechanism. a** ¹⁸O-labeling control experiments. **b** Treat with products with $H_2^{18}O$. **c** Homocoupling byproduct in the absence of arylboronic acids.

significantly increased but the ¹⁸O atom is not associated with the carbonyl group. Furthermore, we found that when the target products (**3a** or **5a**) were directly treated with $H_2^{18}O$, no ¹⁸O-labeling products (**3a-¹⁸O** and **5a-¹⁸O**) were formed (Fig. 5b). Therefore, we envisioned that the oxygen atom exchanging might involve a five-membered ring intermediate, in which the oxygen atom in the phosphonate or the carboxylate may directly react intramolecularly with the alkyne. Evidence of the generated five-membered ring intermediate (**7a** and **7b**) was detected by high-resolution mass spectrometry. Interestingly, without adding arylboronic acids, a 1,3-diene byproduct (**8**, CCDC number: 2202500) can be obtained in 95% yield (Fig. 5c). This would imply that a homocoupling side reaction may occur from direct reductive elimination of high valence gold species.

With data from the mechanism experiments in hand, we proposed the mechanism shown in Fig. 6a. The π-activation of alkyne with the gold(I) catalyst induces the formation of a five-membered ring intermediate (**9**), and this is followed by oxidation with Selectfluor to generate a gold(III) species (**10**). Subsequently, the vinyl-Au(III) intermediate (**10**) would undergo a bimolecular reductive elimination process with arylboronic acids to release the intermediate (**12**) and regenerate the gold(I) catalyst[62,63]. Excess $H_2O$ as a nucleophile, rapidly attacks the highly electrophilic intermediate (**12**) to give rise to the adduct (**13**). Interestingly, the generated intermediate (**13**) will likely to be a nucleophilic species. It can react with Selectfluor reagents to undergo nucleophilic fluorination to directly afford oxo-

arylfluorination product (**3a**) or oxo-arylalkenylation product (**5a**) after H-F elimination in the presence of strong base using tertiary amines generated from the Selectfluor reagents. The formation of oxo-arylfluorination or oxo-arylalkenylation products are highly dependent on using phosphonate or carboxylate as directing groups.

Upon analysis of the reaction mixture by HR-MS (ESI) as shown in Fig. 6b, several kinds of intermediates detected by HR-MS in the oxo-arylfluorination and oxo-arylalkenylation process are proposed, and included the important Au(I) species (**9a, 9b**), the highly electrophilic intermediate (**12a, 12b**) as well as the oxo-arylfluorination intermediate (**14**) in the oxo-arylalkenylation protocol (for details, see the Supplementary Information). These experimental results shed more light on the details of the reaction and further support the possible reaction mechanism shown in Fig. 6. Interestingly, there are two roles for the Selectfluor reagents in these transformations. It can behave as an oxidant to convert Au(I) to Au(III) species and as an electrophilic fluorine donor. Alternatively, for such a complicated system, although a catalytic cycle starting with Au(I)-mediated π-activation was proposed according to our HR-MS results, the first oxidation of Au(I) to Au(III) and then initiation with Au(III)-mediated π-activation cannot be ruled out.

## Discussion

In summary, we have developed a gold-catalyzed four-component relay trifunctionalization reaction of internal alkynes with the assistance of site-directing groups. Both aromatic and aliphatic alkynes can

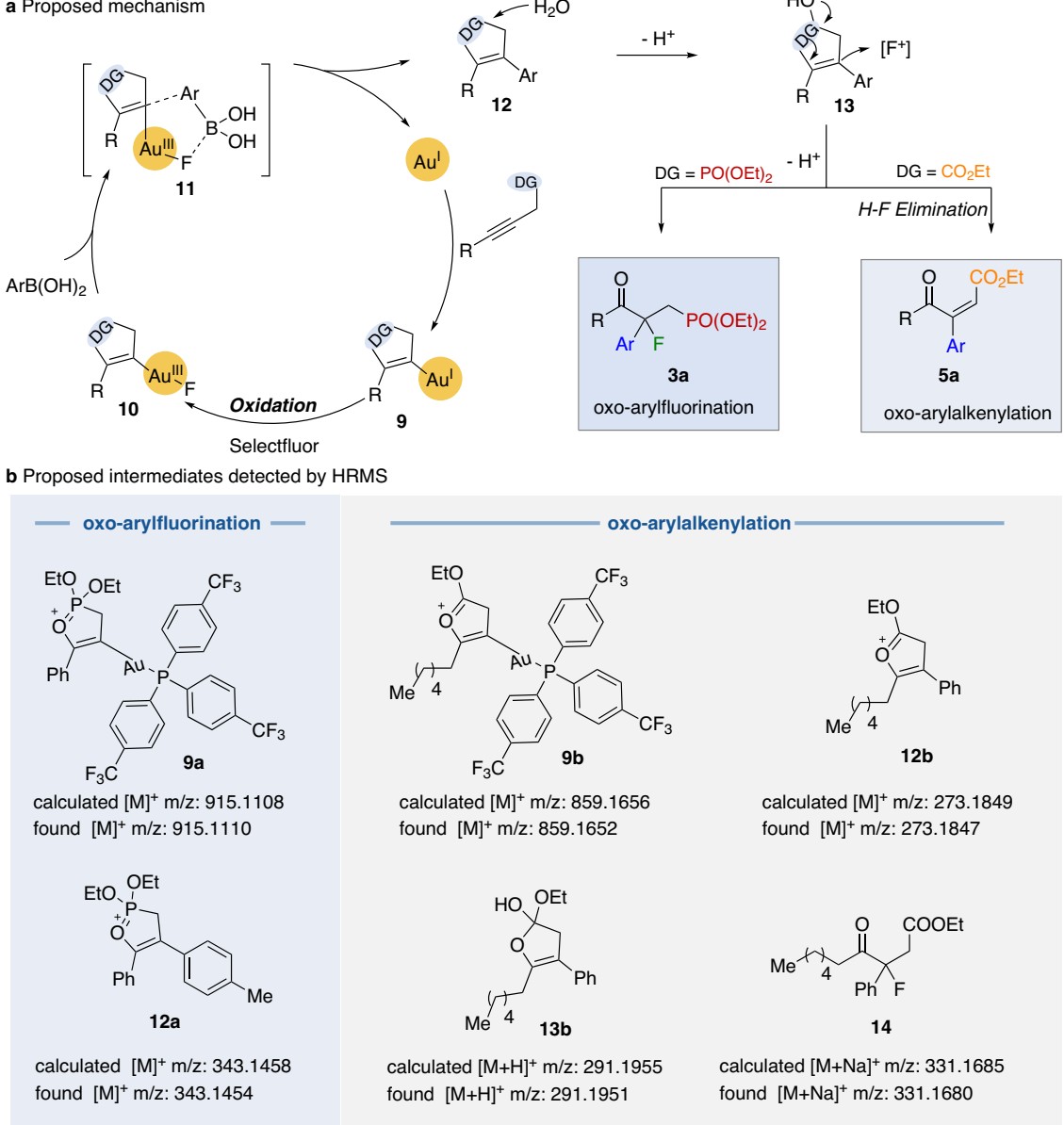

**Fig. 6 | Mechanism proposal. a** Proposed mechanism for this gold-catalyzed four-component multifunctionalization of alkynes. **b** Proposed intermediates detected by HR-MS.

react with arylboronic acids, water and Selectfluor reagents, directly breaking the carbon-carbon triple bonds and forming four new chemical bonds. The different site-directing units hold great promise for the control of the product divergence for oxo-arylfluorination and oxo-arylalkenylation. When the alkyne has a phosphonate as a site-directing group, oxo-arylfluorination can readily occur to furnish a rich library of biologically interesting α,α-disubstituted ketones. At the same time, oxo-arylalkenylation proceeds smoothly when the alkyne contains a carboxylate as the site-directing group and gives rise to a wide range of structurally diverse tri- and tetra-substituted unsaturated ketones. The process has good regioselectivity and functional group compatibility and supports gram-scale preparation and late-stage application of complex alkynes.

## Methods

### General procedure for oxo-arylfluorination of alkynes

To a dried Schlenk tube, (4-CF$_3$Ph)$_3$PAuCl (0.01 mmol, 10 mol%), ArB(OH)$_2$ (0.3 mmol, 3.0 equiv.) and Selectfluor (0.4 mmol, 4.0 equiv.)

are successively added under air atmosphere. Then MeCN (2.0 mL) is added into the tube under stirring conditions. After that, alkynes (0.1 mmol, 1.0 equiv.) and water (0.2 mmol, 2.0 equiv.) are added by microinjector under air atmosphere. The resulting reaction mixture is heated at 50 °C for 12 h. When the reaction is finished (monitored by TLC), the reaction mixture is cooled to room temperature, and extracted by ethyl acetate, dried over anhydrous Na$_2$SO$_4$. The combined organic layer is concentrated in vacuo. The resulting residue is purified by column chromatography on silica gel (eluent: petroleum ether/ethyl acetate) to give the final product **3**.

### General procedure for oxo-arylalkenylation of alkynes

To a dried Schlenk tube, (4-CF$_3$Ph)$_3$PAuCl (0.01 mmol, 10 mol%), ArB(OH)$_2$ (0.3 mmol, 3.0 equiv.), Selectfluor (0.25 mmol, 2.5 equiv.) and MeCN (2 mL) are added successively under air atmosphere. After that, the corresponding alkynes (0.1 mmol, 1.0 equiv.) and water (0.2 mmol, 2.0 equiv.) are added by microinjector under air atmosphere. The resulting reaction mixture is heated at 50 °C for 12 h and

then cooled to room temperature. The reaction mixture is cooled to room temperature, and extracted by ethyl acetate, dried over anhydrous $Na_2SO_4$. The combined organic layer is concentrated in vacuo. The resulting residue is purified by column chromatography on silica gel (eluent: petroleum ether/ethyl acetate) to give the final product **5**.

## Data availability

Crystallographic data for the structures reported in this Article have been deposited at the Cambridge Crystallographic Data Centre, under deposition numbers CCDC 2202498 (**3n**), CCDC 2202499 (**5u**) and CCDC 2202500 (**8**). Copies of the data can be obtained free of charge via https://www.ccdc.cam.ac.uk/structures/. Data related to materials and methods, optimization of conditions, experimental procedures, mechanistic experiments, and spectra are provided in the Supplementary Information. All data are available from the corresponding authors upon request.

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

## Acknowledgements

We thank the National Key Research and Development Program of China (2022YFA1503200 and 2021YFC2101901 to J.X.), National Natural Science Foundation of China (22122103 and 21971108 to J.X., 22101130 to J.H., 21971111 and 22271144 to C.Z.), Fundamental Research Funds for the Central Universities (020514380304, 020514380252 and 020514380272 to J.X.) for financial support. J.H. acknowledges the supporting of Xiaomi foundation. All theoretical calculations were performed at the High-Performance Computing Center (HPCC) of Nanjing University. Chenglong Ji, Yantao Li, and Nian Li at Nanjing University are warmly acknowledged for their reproduction of the experimental procedures for products **3a**, **3y**, and **5a**.

## Author contributions

J.X. and S.F. conceived the work and designed the experiments. S.F., J.H. and W.L. performed the experiments and analyzed the experimental data and discussed with C.Z., and J.X. co-wrote the manuscript with input from all the other authors.

## Competing interests

The authors declare no competing interests.
