## [Peer Review File · Nature Communications]

Gold-Catalyzed Four-Component Multifunctionalization of AlkynesReviewers' Comments:

Reviewer #1:

Remarks to the Author:

Xie and co-workers developed a gold-catalyzed multifunctionalization of alkynes tuned by site-directing groups. Gold-catalyzed four-component reactions of alkynes, arylboronic acids, water, and selectfluor have been achieved to afford the oxo-arylfluorination or oxo-arylalkenylation depending on the directing group of alkynes. The reaction divergence has been controlled by site-directing functional groups in the alkynes; i. e. when phosphonate group is used it favors the oxo-arylfluorination while the carboxylate motif gives oxo-arylalkenylation products. This method provides α,α -disubstituted ketones and tri- or tetra-substituted unsaturated ketones with excellent chemo-, regio- and stereoselectivity. Moreover, the gram scale synthesis and late-stage functionalization of complex alkynes have been shown. From the mechanistic viewpoint, O18 labeling experiments have been carried out to show oxygen incorporation. Further, key intermediates have been detected by mass spectrometric analysis. Moreover, in the case of oxo-arylalkenylation reactions, anti-periplanar staggered conformation has been proposed to rationalize the stereochemistry of alkene in the products. The following points need to address before the publications

1. Can authors comment on the scope with respect to electron-donating aryl boronic acids? Similarly, what about ortho-substituted aryl-boronic acid? It is fine even if the reaction does not work but this should be indicated in the manuscript.
2. How do terminal alkynes behave under standard reaction conditions?
3. In figure 5a – in the second O18 labeling experiment, 2a should be 4a.
4. Indeed, the reaction of alkynes in Au(I)/Au(III) catalysis are rare. However, there exist some examples. I encourage the authors to carefully check the literature and cite the reports on the reaction of alkynes under Au(I)/Au(III) catalysis.

In summary, the field of Au(I)/Au(III) catalysis is emerging and this example provides an interesting addition to the growing field. The four-component reaction is indeed challenging and here the authors have realized the interesting reactivity. Hence, I strongly support its publication once the above minors points have been addressed.

Reviewer #2:

Remarks to the Author:

In this manuscript, Xie's group reports a reaction protocol for gold-catalyzed oxo-arylfluorination or oxo-arylalkenylation of alkyne. In this reaction, the author achieved divergence synthesis by using phosphonate or carboxylate as directing groups. Interestingly, selectfluor is used as the oxidant and a fluorinating reagent in the phosphonate-directing case. This work has been performed in good quality with abundant boronic acid and alkyne substrates. Gram scale synthesis and complex substrates are achieved. The water-18O experiments were investigated and provide the five-membered ring as the key intermediate. Due to the synthetically useful of this work, this is a good paper that fits the general interest and quality requirement for Nature Communications.

Some minor suggestions for revisions:

- Proposed mechanism: Gold chloride is used as the catalyst. Is the mechanism more likely to start from gold oxidant to Au(III) instead of gold(I) activation?
- NMR has Solvent: 3cc
- NMR needs to purify: 1aa, 4t
- Figure 5 Bar chart should have a better resolution.
- The related papers on alkyne activation reactions such as Chem, 2020, 6, 1420; J. Am. Chem. Soc. 2014, 136, 14654. should be cited.

Reviewer #3:

Remarks to the Author:

The most interesting part of this manuscript is the phosphate directed functionalization of alkynes to afford α -aryl- α -fluoroketones, although the products are somewhat esoteric and many 'entries' are redundant.

Another interesting aspect is the mechanistic switch afforded by carboxylalkyl direction. That is sort of neat, and it makes a lot of sense in context. Once again there is some redundancy in the entries, but the other explorations are thorough.

The quality of the English is uneven and sometimes nonidiomatic, and the overall style of the paper is a bit boring. The authors must know that they are setting their manuscript a sail in a stormy sea when they have not subjected it to a proofreading for idiomatic usage. I do not have time to correct it all, so I suggest they do what they should have done at the outset and do it right.

Reviewer #1 (Remarks to the Author):

Xie and co-workers developed a gold-catalyzed multifunctionalization of alkynes tuned by site-directing groups. Gold-catalyzed four-component reactions of alkynes, arylboronic acids, water, and selectfluor have been achieved to afford the oxo-arylfluorination or oxo-arylalkenylation depending on the directing group of alkynes. The reaction divergence has been controlled by site-directing functional groups in the alkynes; i. e. when phosphonate group is used it favors the oxo-arylfluorination while the carboxylate motif gives oxo-arylalkenylation products. This method provides α,α -disubstituted ketones and tri- or tetra-substituted unsaturated ketones with excellent chemo-, regio- and stereoselectivity. Moreover, the gram scale synthesis and late-stage functionalization of complex alkynes have been shown. From the mechanistic viewpoint, O18 labeling experiments have been carried out to show oxygen incorporation. Further, key intermediates have been detected by mass spectrometric analysis. Moreover, in the case of oxo-arylalkenylation reactions, anti-periplanar staggered conformation has been proposed to rationalize the stereochemistry of alkene in the products.

Thank you very much for your positive evaluation to the synthetic values of our new synthetic methodology. Much appreciated!

The following points need to address before the publications

1. Can authors comment on the scope with respect to electron-donating aryl boronic acids? Similarly, what about ortho-substituted aryl-boronic acid? It is fine even if the reaction does not work but this should be indicated in the manuscript.

Answer: Thank you very much for your important suggestions, and we will show more details about electron-donating aryl boronic acids and ortho-substituted aryl-boronic acid as follow: "Notably, the substituents on the ortho-position of the phenyl rings such as 2-tolylboronic acid, make the oxo-arylfluorination

process reluctant, probably due to the steric influence. Besides, it is found that when the strongly electron-donating 4-methoxy substituted phenylboronic acid was subjected to the standard reaction conditions, no expected reaction occurred.”

2. How do terminal alkynes behave under standard reaction conditions?

Answer: Thank you very much for your helpful comments. We tried some terminal alkynes under similar reaction condition. However, there were no desired products.

Investigation of terminal alkynes

no desired product

no desired product

no desired product

3. In figure 5a – in the second O18 labeling experiment, 2a should be 4a.

Answer: Thank you very much for your important suggestions, and the correction has been applied.

4. Indeed, the reaction of alkynes in Au(I)/Au(III) catalysis are rare. However, there exist some examples. I encourage the authors to carefully check the literature and cite the reports on the reaction of alkynes under Au(I)/Au(III) catalysis.

Answer: Thank you very much for your important suggestions. We check the area of gold-catalyzed alkynes transformation again, and cite some related work.

42. Tathe, A. G., Saswade, S. S. & Patil, N. T. Gold-catalyzed multicomponent reactions. *Org. Chem. Front.* <https://doi.org/10.1039/D3QO00272A> (2023).
43. Peng, H. et al. Gold-Catalyzed Oxidative Cross-Coupling of Terminal Alkynes: Selective Synthesis of Unsymmetrical 1,3-Diynes. *J. Am. Chem. Soc.* **136**, 13174–13177 (2014).
44. Joost, M. et al. Facile Oxidative Addition of Aryl Iodides to Gold(I) by Ligand Design: Bending Turns on Reactivity. *J. Am. Chem. Soc.* **136**, 14654–14657 (2014).
45. Yuan, T. et al. Regioselective Crossed Aldol Reactions under Mild Conditions via Synergistic Gold-Iron Catalysis. *Chem* **6**, 1420–1431 (2020).
46. Liu, K., Li, T., Liu, D.-Y., Li, W., Han, J., Zhu, C., & Xie, J. Dinuclear Gold-Catalyzed C-H Bond Functionalization of Cyclopropenes, *Sci. China Chem.* **64**, 1958–1963 (2021)
47. Guenther, J., et al. Activation of Aryl Halides at Gold(I): Practical Synthesis of (P,C) Cyclometalated Gold(III) Complexes. *J. Am. Chem. Soc.* **136**, 1778–1781 (2014).
48. Bhojare, V. W., Carrizo, E. D. S., Chintawar, C. C., Gandon, V., & Patil, N. T. Gold-Catalyzed Heck Reaction. *J. Am. Chem. Soc.* DOI: 10.1021/jacs.3c02544 (2023).

In summary, the field of Au(I)/Au(III) catalysis is emerging and this example provides an interesting addition to the growing field. The four-component reaction is indeed challenging and here the authors have realized the interesting reactivity. Hence, I strongly support its publication once the above minors points have been addressed.

Reviewer #2 (Remarks to the Author):

In this manuscript, Xie's group reports a reaction protocol for gold-catalyzed oxo-arylfuorination or oxo-arylalkenylation of alkyne. In this reaction, the author achieved divergence synthesis by using phosphonate or carboxylate as directing groups. Interestingly, selectfluor is used as the oxidant and a fluorinating reagent in the phosphonate-directing case. This work has been performed in good quality with abundant boronic acid and alkyne substrates. Gram scale synthesis and complex substrates are achieved. The water-18O experiments were investigated and provide the five-membered ring as the key intermediate. Due to the synthetically useful of this work, this is a good paper that fits the general interest and quality requirement for Nature Communications.

Thank you very much for your positive evaluation to the synthetic values of our new synthetic methodology. Much appreciated!

Some minor suggestions for revisions:

- Proposed mechanism: Gold chloride is used as the catalyst. Is the mechanism more likely to start from gold oxidant to Au(III) instead of gold(I) activation?

Answer: Thank you very much for the helpful discussion. We also have considered this. However, from our HRMS analysis data of the reaction mixture, gold(I)-activated intermediate was detected. According to your suggestion, one new sentence was added in the revised manuscript as "Alternatively, for such a complicated system, although a catalytic cycle starting with Au(I)-mediated π -activation was proposed according to our HRMS results, the first oxidation of Au(I) to Au(III) and then initiation with Au(III)-mediated π -activation cannot be ruled out."

- NMR has Solvent: 3cc

- NMR needs to purify: 1aa, 4t

Answer: Thanks for your sincere advice. We have done the purification.

- Figure 5 Bar chart should have a better resolution.

Answer: Thanks for your reminder. We have modified it in the revised manuscript.

- The related papers on alkyne activation reactions such as *Chem*, 2020, 6, 1420; *J. Am. Chem. Soc.* 2014, 136, 14654. should be cited.

Answer: Thank you very much for your helpful comments. According to your suggestion, we have added the references in the manuscript as following:

42. Peng, H. et al. Gold-Catalyzed Oxidative Cross-Coupling of Terminal Alkynes: Selective Synthesis of Unsymmetrical 1,3-Diynes. *J. Am. Chem. Soc.* 136, 13174–13177 (2014).
43. Joost, M. et al. Facile Oxidative Addition of Aryl Iodides to Gold(I) by Ligand Design: Bending Turns on Reactivity. *J. Am. Chem. Soc.* 136, 14654–14657 (2014).
44. Yuan, T. et al. Regioselective Crossed Aldol Reactions under Mild Conditions via Synergistic Gold-Iron Catalysis. *Chem.* 6, 1420–1431 (2020).

Reviewer #3 (Remarks to the Author):

The most interesting part of this manuscript is the phosphate directed functionalization of alkynes to afford α -aryl- α -fluoroketones, although the

products are somewhat esoteric and many 'entries' are redundant.

Answer: Thanks for your sincere advice and suggest it is an interesting work.

Another interesting aspect is the mechanistic switch afforded by carboxylalkyl direction. That is sort of neat, and it makes a lot of sense in context. Once again there is some redundancy in the entries, but the other explorations are thorough.

Answer: Thanks for your sincere advice. To further showcase the functional group tolerance of this protocol, a series of substrates are employed.

The quality of the English is uneven and sometimes nonidiomatic, and the overall style of the paper is a bit boring. The authors must know that they are setting their manuscript a sail in a stormy sea when they have not subjected it to a proofreading for idiomatic usage. I do not have time to correct it all, so I suggest they do what they should have done at the outset and do it right.

Answer: Thank you very much for your helpful suggestions and kind support of our work. We are full agreement with you. As the non-English native speakers, although we have tried to write carefully, there are several problems. According to your suggestion, we have asked Prof. Dr. George W. A. Milne (a retired research chemist at the National Institutes of Health in Bethesda, MD) to help us polish the language. Thank you very much for your kind support and share us these very useful suggestions.

Reviewers' Comments:

Reviewer #1:

Remarks to the Author:

All the concerns have been addressed in the revision. Hence I recommend the publication of this manuscript in its present form.

Reviewer #2:

Remarks to the Author:

In this revision, the authors provide good explanation/addition in addressing the previously raised concerns. This reviewer therefore supports the publication of this work as is.

REVIEWERS' COMMENTS

Reviewer #1 (Remarks to the Author):

All the concerns have been addressed in the revision. Hence I recommend the publication of this manuscript in its present form.

Answer: Thanks for your comments.

Reviewer #2 (Remarks to the Author):

In this revision, the authors provide good explanation/addition in addressing the previously raised concerns. This reviewer therefore supports the publication of this work as is.

Answer: Thanks for your comments.